# Osteonecrosis Related to Steroid and Alcohol Use—An Update on Pathogenesis

**DOI:** 10.3390/healthcare11131846

**Published:** 2023-06-26

**Authors:** Wojciech Konarski, Tomasz Poboży, Klaudia Konarska, Andrzej Śliwczyński, Ireneusz Kotela, Martyna Hordowicz, Jan Krakowiak

**Affiliations:** 1Department of Orthopaedic Surgery, Ciechanów Hospital, 06-400 Ciechanów, Poland; tomasz.pobozy@onet.pl; 2Medical Rehabilitation Center, Sobieskiego 47D, 05-120 Legionowo, Poland; klodiii87@gmail.com; 3Social Medicine Institute, Department of Social and Preventive Medicine, Medical University of Lodz, 90-647 Lodz, Poland; andrzej.sliwczynski.ahe@gmail.com (A.Ś.); jan.krakowiak@umed.lodz.pl (J.K.); 4Department of Orthopedic Surgery and Traumatology, Central Research Hospital of Ministry of Interior, Wołoska 137, 02-507 Warsaw, Poland; ikotela@op.pl; 5Department of Psychiatry, Independent Regional Complex of Public Psychiatric Health Care Facilities in Warsaw, 00-665 Warsaw, Poland; m.hordowicz@gmail.com

**Keywords:** osteonecrosis, avascular necrosis, femoral head, alcohol, alcohol abuse, steroids, ethanol, corticosteroids, disability, pathogenesis

## Abstract

Osteonecrosis (ON) is usually a progressive disease that negatively affects the quality of life and leads to significant disability. Most patients are aged 30–50 years and require multiple surgical interventions throughout their lifetime. In non-traumatic ON, alcohol abuse and corticosteroids are involved in up to 80% of cases. This narrative review aims to summarize data on their impact on healthy bone and the development of pathophysiological processes leading to ON development. We conducted EMBASE and MEDLINE database reviews to identify relevant research. We found that for both agents, the risk was time and dose-dependent. ON in alcohol and steroid use shared many pathogenetic mechanisms leading to the development of necrosis, including increased adipogenesis, the induction of chronic inflammation, vascular alterations, and impaired bone-cell differentiation. Because both alcohol and steroid use are modifiable factors, both general physicians and orthopedic surgeons should encourage patients to limit ethanol intake and avoid corticosteroid overuse. In the presence of ON, because both alcohol- and steroid-induced disease tend to be multifocal, addiction treatment and limiting steroid use are justified.

## 1. Introduction

Osteonecrosis (ON), often referred to as avascular necrosis (AVN) or sterile necrosis, consists of necrotic changes in the bone related to either the sudden, permanent, or temporary depletion of the blood supply [1,2]. Usually, the sites affected have limited collateral circulation. The most common location is the femoral head [3].

In the US and Europe, 2–10% of total hip arthroplasties are performed due to osteonecrosis of the femoral head (ONFH), but in Asia, this percentage might be as high as 60% [4,5]. Up to 20,000 patients in the US are diagnosed with ONFH each year [1,6]. The yearly case count in Germany alone is between 5000 and 7000 [6]. Patients with femoral head involvement usually present with pain in the groin area. Bilateral involvement is found in 40–70% of the cases, with the contralateral side being asymptomatic [7].

ON is usually progressive, negatively affecting the quality of life and functioning and finally leading to joint replacement surgeries and significant disability [1]. Most patients are 30–50 years, and because of their young age, joint preserving techniques are often implemented as these individuals would require multiple surgical interventions throughout their lifetime [1,4]. To date, there is no consensus regarding the best treatment for ON [1].

Multiple risk factors, toxins, and events may precipitate ON. In non-traumatic ON, alcohol abuse and corticosteroids are involved in the majority of cases [2]. Systemic lupus erythematosus (SLE), solid organ transplantation, hematologic diseases, dyslipidemias, obesity, and many more are also considered risk factors [1,2,3]. Even though extensive research has been conducted to date to elucidate the pathogenetic mechanisms for ON, this topic remains largely unexplored [1]. Most likely, one pathomechanism is not sufficient to trigger the disease.

The aim of this paper is to summarize the current knowledge on the two most prevalent factors leading to bone aseptic necrosis unrelated to trauma—steroid- and ethanol-induced ON.

## 2. Materials and Methods

This review aims to summarize current concepts regarding pathophysiological processes leading to ON development in alcohol abuse and steroid treatment. We synthesized our findings in the form of a narrative review. Since systematic reviews with metanalyses were created to answer narrowly focused questions, that methodology was assessed as being unsuitable for a review designed to describe and critically analyze the available knowledge in a wider context [8]. Our work aimed to explore a broader scope of research works published than systematic reviews allow. The key objective was to accomplish the purpose of narrative reviews: to create an interpretative understanding of the current landscape of knowledge regarding ON and its relationship with alcohol and steroid use [8,9].

Information on the impact of alcohol and corticosteroids on healthy bone tissue was also included, as it is directly linked to the development of osteonecrosis and other bone pathologies. This review’s purpose was not to compare individual studies.

We searched PubMed and EMBASE databases using a combination of the phrases “osteonecrosis”, “avascular necrosis”, and “osteonecrosis of the femoral head” with “alcohol”, “alcohol abuse”, “alcoholism”, “ethanol”, or “steroids”, “corticosteroids”, including papers from 1975 until March 2023. All keywords used for the literature search are shown in Table 1. Papers in English, Polish and Spanish languages were included. Other languages were considered for inclusion in the case that the abstracts were informative enough and available in any of the above languages.

We adapted broad inclusion criteria while screening the results to allow for the incorporation of a wide array of data. We also screened the references to expand the possibility of including relevant literature. The inclusion criteria were as follows:Papers discussing the pathophysiology of osteonecrosis in relation to steroid treatment AND/OR alcohol abuse;Research regarding the impact of exogenous steroids and alcohol on bone tissue and its physiology, including its structure, function, cell development and tissue resistance, and other aspects;Types of papers: original research, including human and animal research, review papers (including systematic reviews and meta-analyses), recommendations, and guidelines. Only peer-reviewed sources were suitable for this review.

We excluded studies that focused on in vitro research, general research, and reviews discussing the side effects of steroids and consequences of alcohol abuse that did not discuss the impact on the bone and ON. Editorials, opinions, and expert commentaries were not considered suitable for this review. The final selection of papers was based on a critical assessment by the authors, taking into consideration the papers’ relevance, significance and contribution to shaping the narrative framework.

## 3. Results

Osteonecrosis has a multifactorial etiology, resulting from an intricate interplay of various predispositions that can be inherent to an individual’s traits or influenced by external factors. The latter encompass the consumption of ethanol or specific medications, including steroids, which can also interact with intrinsic predisposing characteristics such as diseases (e.g., diabetes mellitus and certain hematologic conditions) and genetic makeup. These distinct factors collectively contribute to the same pathogenetic mechanism, namely, alterations in the bone’s blood supply [10]. In the presence of numerous risk factors, the likelihood of osteonecrosis development may be amplified beyond a mere summation of the risks associated with each individual factor. Table 2 provides a summary of the most common factors contributing to the development of osteonecrosis.

The femoral head is a frequent site of ON because of very its limited collateral blood flow [4]. Because its joint surfaces are highly dependent on narrow, terminal blood vessels, they are more prone to necrosis than other bones. When blood flow is disrupted, ischemia develops, eventually leading to necrotic cell death [1,10]. The histopathologic picture of osteonecrosis consists of pycnotic nuclei or dispersed empty lacunae situated in the bone trabeculae, surrounded by necrotic cells [11]. The weakening of the bone structure results in an accumulation of microfractures over time. Consequently, more apparent fractures appear, and the joint surface becomes deformed. In the final stage, the joint eventually collapses [10].

The sudden disruption of blood flow is the leading cause of ON in trauma. In ON related to steroid and alcohol use, blood flow gradually diminishes in the long term due to the accumulation of the implications of various changes that limit the ability of the bone tissue to heal effectively [4]. Additionally, several gene polymorphisms identified to date have been hypothesized to influence ON development in individuals exposed to steroids or alcohol. Though a considerable amount of research has been conducted over the last decades, there is no consensus regarding the pathophysiological processes of the development of non-traumatic ON. One pathomechanism is most likely insufficient to trigger the disease [11]. Below, we discuss the nuances regarding alcohol and steroid impact on a healthy bone and the pathogenesis of ON resulting from exposure to them.

### 3.1. Alcohol and Osteonecrosis

Alcohol is considered the oldest intoxicant invented by humans [12]. It is the most abused legal drug in both the US and Europe, with numerous health consequences and high social and economic burdens [12,13].

Alcohol abuse is related to increased morbidity and mortality in more than 60 conditions, including different types of cancer, liver cirrhosis, cardiovascular disease, cognitive impairment, and adverse changes in behavior [13]. Prior to the outbreak of the COVID-19 pandemic, one in five Europeans was addicted to alcohol, with the prevalence ranging from 0,1 to 6.6% [13,14]. Studies conducted during the pandemic demonstrated an increased frequency and quantity of alcohol consumption in many European countries, including the United Kingdom, Germany, and Poland [15,16,17]. The consequences of abuse for public health will take an even greater toll in the upcoming years.

The effects of ethanol are also reflected in the bone. Numerous pathophysiological processes are triggered by excessive alcohol consumption, leading to an increased risk of ON development in drinkers. The Association Research Circulation Osseous classification (ARCO) set criteria for alcohol-related osteonecrosis in 2017 [18]. They involve a weekly ethanol consumption of >320 g per week for more than six months, and the disease must occur within a year after consuming that amount. In addition, individuals with ethanol-induced ON should not have confounding risk factors present [18].

### 3.2. Epidemiology of ON in Alcohol Abuse

Epidemiological studies demonstrate that alcohol is one of the most prevalent risk factors for osteonecrosis, present in approximately 20–45% of individuals with this condition [19,20,21]. The incidence of aseptic necrosis in regular drinkers was found to be as high as 5.3% [22]. Multifocal disease was found in 6% [23]. Alcohol intake had a dose-dependent impact on ON occurrence. In a study by Tsai et al. among the Taiwanese population [20], it was identified as a predominant cause of ONFH in 45.2% of 1153 individuals. Their average consumption was 86 g. Continued exposure to ethanol leads to inferior outcomes of ON treatment, with 5-year survival rates after core decompression being the lowest for alcohol-associated ONFH (22.1% vs. 44.1% in idiopathic cases and 30.1% in steroid-induced ONFH) [20]. Another study conducted in China also indicated that alcohol was the most common cause of ONFH (685/1844, 37.5%), with steroids ranking as second (495/1844, 26.84%) and traumatic ONFH as third (290/1844, 15.73%) [24]. However, in a smaller Japanese report by Ikeuchu et al. [25], the proportions were reversed, with steroids being linked to 47.4% of cases (135/244). Despite that, the proportion of alcohol-induced ONFH was high (87/244, 30.5%). In 4.9% (14/244) cases, steroid- and alcohol use were related to bone necrosis. Similar findings were described by Sato et al. [26]. In their study, 3264 individuals were newly diagnosed with ONFH. The annual incidence of ONFH was 3.0/100,000. The most prevalent precipitating factors were steroid use (39%), alcohol (30%), or both (4%). A total of 27% of cases were idiopathic [26]. In a small study involving 13 HIV patients from Korea, smoking, steroid-, and alcohol use were the key risk factors for ONFH (in 10, 5, and 4 patients, respectively) [27]. In conclusion, alcohol-induced ON is one of the two most common causes of non-traumatic ON, with a poor prognosis in terms of treatment outcomes.

#### 3.2.1. Alcohol Effects on Bone Health

Bone connective tissue is a living structure containing a mineralized cellular matrix and cells, providing a structural framework for our bodies [28]. Alcohol interferes with both cellular and acellular bone components, directly and indirectly. Among the indirect effects, malnutrition, low body weight, low vitamin D levels, and hormonal imbalance may be enumerated [29]. Thiamine (vitamin B1), the most deficient vitamin in heavy drinkers, might also contribute to the deterioration of bone health through its impact on osteoclasts’ differentiation and genesis. B1 vitamin deficits negatively impact bone formation and its mechanical resistance, which increases susceptibility to ON [30]. Riboflavin (B2) deficiency, which is also connected to alcohol abuse, was also shown to have a protective effect against ON by preventing vascular damage [31].

During adulthood, bone tissue undergoes constant remodeling, in which the bone is broken down and rebuilt. A complete cycle takes about 4–6 months [29,32]. A decrease in bone buildup with an increase in resorption seen in alcohol abusers results in lower bone mineral density (BMD), which reduces its resistance [33]. That makes it prone to fractures and eventually leads to osteopenia and, finally, osteoporosis. Alcohol induces osteopenia through decreases both in osteoblast activity and hormone-dependent mechanisms [29,34]. These effects are mediated through hormones that regulate calcium concentration, of which the bone is the primary storage location [29]. Parathormone (PTH) secretion increases post-acute drinking, leading to an increased level of calcium [34]. Chronic alcohol consumption, on the other hand, decreases PTH secretion. It also alters vitamin D metabolism. The 25-hydroxyvitamin D levels are decreased in alcohol users, in part because of low 25-hydroxylase activity and low intestinal absorption [35]. Cirrhotic patients, due to hepatic protein synthesis decreases, also have low vitamin D binding protein levels, which lead to low concentrations of 1,25-hydroxy vitamin D level [29]. Vitamin D deficit decreases calcium absorption in the intestines and causes phosphate deficiency [29,35]. It has been demonstrated that alcohol-induced osteopenia is not strictly determined by the amount of ethanol ingested, but is mainly driven by malnutrition [35]. Alcohol-related endocrine disruption affects other hormones as well. Insulin growth factor-1 (IGF-1) and reproductive hormone production and metabolism are disrupted in both men and women [34].

At the level of the bone tissue itself, numerous biomechanical processes are implicated in the alcohol-related weakening of bone strength and resistance. Research has demonstrated that ethanol impacts the precursor mesenchymal cells’ potential to differentiate into the osteoblastic lineage [5]. Alkaline phosphatase activity and the expression of osteocalcin were reduced. Both are markers of bone mineralization, and diminished values were found in individuals with fractures and osteoporosis [36,37]. As a consequence, the bone’s formation and mineralization, as well as its healing capacity, becomes significantly reduced [29]. These processes intensify with more prolonged exposure and ethanol concentration [38]. An exception was found in elderly women, where moderate drinking was factually related to an increase in BMD [29]. Interestingly, fractures in alcohol-dependent individuals tended to occur with BMD above the fracture threshold. Peris et al. observed that while more than 1 in 3 patients suffered a vertebral fracture, only 6.5% had a BMD below the fracture threshold [39].

#### 3.2.2. Pathogenesis of Ethanol-Related Osteonecrosis

The relationship between ON and alcohol consumption is both time and dose-dependent. Increased odds of developing that condition were found in occasional drinkers (relative odds = 3.2; 95%CI 1.1–9.2) but are increased fourfold in regular consumers (relative odds = 13.1, 95%CI; 4.1–42.5), with statistically significant dose dependence [40]. Other studies confirm the dose-dependent relationship, with relative risks (RR) of ON ranging from 3.3 in individuals consuming 400 mL a week or less to 17.9 when the intake exceeds 1000 mL weekly. With the concurrent use of steroids, the risk of ON is even higher [19].

The duration of alcohol use correlates with the odds of developing ON. Nonetheless, even short-term drinking leads to histological changes in the bone, making it more susceptible. A study involving rats conducted by Okazaki et al. [41] demonstrated that four weeks of daily consumption of a 5% alcohol liquid diet prompted the appearance of necrotic areas surrounded by abnormal, appositional bone formation. Still, evident changes during the histological examination were visible in animals sacrificed as early as after the first week of the experiment [41]. They also observed alterations in lipid metabolism and elevated liver enzymes. These changes were more intense in the following weeks. The authors concluded that although most ONFH cases are observed in long-term drinkers, immoderate ethanol consumption might cause alcohol-induced osteonecrosis much earlier than previously thought [41].

Ethanol-induced osteopenia and osteoporosis are among the causes of ON in alcohol users. Reduced bone density is strictly correlated with both trauma-related and non-traumatic ONFH; pre-collapse stages and more advanced ON were related to a 5-fold increase in the risk of osteoporosis [10]. Accumulating microfractures over time with reduced remodeling capacity represents another pathogenic mechanism of ethanol-induced ON [10,29]. The induction of bone resorption further impacts bone resistance, making it even more prone to fracture and collapse. Research suggests that ethanol also decreases bone resistance by limiting osteoblasts’ ability to develop into osteocytes and inducing their apoptosis [21,29,34].

Another aspect that might contribute to ON through decreased joint stability and an increased propensity for trauma is low muscle mass and strength, which is present in both alcohol users with and without liver damage. Skeletal muscles provide both joint stabilization and protection. Excessive alcohol consumption leads to alterations in the body composition, with a higher fat-to-lean muscle mass ratio in comparison with non-abusers [28]. Altered protein synthesis, malabsorption, and malnutrition in liver cirrhosis lead to sarcopenia, as well [29].

In addition to ethanol’s impact on bone formation and resistance, another well-studied mechanism for ON is related to alcohol-induced lipid metabolism disorders, which lead to increased adipogenesis [38]. In rabbits fed with ethanol 10 mL/kg/day for up to six months, fatty infiltrates were found in both the livers and the bone marrow of animals upon histological examination [42]. These cells filled up with triglycerides and, over time, became pycnotic. As adipocytes become hypertrophic, they compress blood vessels, leading to their occlusion and small vein compression. The cessation of blood flow continuity results in ischemia, cell death, and intravascular coagulation [18]. In addition, the excessive growth of adipocytes suppresses hematopoiesis in the subchondral area [42]. Lipid accumulation in the bone tissue during the failed reparative process leads to the saponification of necrotic bone marrow [38,43]. Also, the inflammatory process evoked by necrosis, instead of removing necrotic cells and promoting repair through the recruitment of immune cells as in the acute state, changes into a chronic process due to alcohol and its metabolites [44].

Some gene polymorphisms have been found to modulate the risk of ethanol-induced ON. MIR31HG gene polymorphisms (such as rs10965059 and rs10965064), which play a role in osteogenic differentiation and osteogenesis, were related to a reduced risk of alcoholic ON in the Chinese population [44]. Some Rab40c gene polymorphisms, on the other hand, increase risk. CARMEN (Cardiac Mesoderm Enhancer-Associated Non-Coding RNA) variants might decrease the risk of ON of the femoral head in alcohol use in Chinese individuals [45]. In the Chinese population, apolipoprotein B (ApoB) variants were associated with a reduced risk of necrosis in alcohol abusers [46].

In summary, the pathogenesis of ON in alcohol users is multifactorial, involving both its negative impact on bone health, osteogenesis, and remodeling, and inducing chronic inflammation, disorders in the lipid metabolism, decreased arterial and venal blood flow, and intravascular coagulation. These processes depend on both the exposure time and the toxin dose.

### 3.3. Steroid Use and Osteonecrosis

The first use of corticosteroids dates back to the 1950s [47]. Since then, these agents have been widely used to relieve various ailments due to their anti-inflammatory and immunomodulatory properties [48]. Nevertheless, initial enthusiasm was tempered by emerging evidence of the deleterious effects associated with long-term systemic exposure [47]. Currently, steroid-induced osteoporosis is responsible for a quarter of secondary osteoporosis cases, the second-most-common cause in both men and women [48,49,50]. Steroid use is also a pivotal risk factor for the development of osteonecrosis of the bone, which is responsible for 3–40% of non-traumatic cases of ON [10,19,20,21,51,52]. The administration of these drugs may increase the risk of ON by up to 20 times [5]. It was also found that steroid use in some conditions is associated with multifocal disease, mainly rheumatic diseases—including SLE, malignancies (such as acute lymphoblastic leukemia and non-Hodgkin lymphoma), and pulmonary disease [51,53].

Before the 1980s, the evidence for an association between steroids and ON was mixed. Since then, a mounting amount of research has confirmed a link between the two, and that it is, simplistically put, dose-dependent. Short-term high-dose therapy and chronic use are related to the highest risk [51,54,55]. Nevertheless, even a single dose of steroids is capable of inducing ON in animal models [56,57], though there have been only a few such case reports in humans [51]. Between 9% and 40% of patients administered corticosteroids will eventually develop ON. The risk is considerably higher in those receiving long-term therapy, especially with doses exceeding the 15–20 mg/day range [19,51,55]. The discontinuation of corticosteroid treatment does not entirely reverse the pathogenetic processes responsible for ON [58].

#### 3.3.1. Steroids’ Role in Bone Health and Disease

Corticosteroids evoke their actions mainly intracellularly, through nuclear receptors, which work as transcription factors [50]. Nevertheless, some are mediated through non-genomic mechanisms. The physiological role of endogenous cortisol in bone health has yet to be fully elucidated. Varying concentrations of cortisol during the day and with aging complicate research in this area [59]. Physiological levels of endogenous steroids are necessary for bone mass maintenance [50,59]. It is clear, however, that excess endogenous (as in the Cushing syndrome) and therapeutic steroids, as well as their deficiency (as seen in Addison’s syndrome), may seriously impact bone homeostasis [50,51,53,55,59].

Individuals with both Addison’s disease and Cushing syndrome (including iatrogenic) have an increased propensity to bone fractures [50,59,60]. Physiological levels of steroids stimulate osteoblast function and proliferation, which translates into an improvement in bone mass [50,60]. In the presence of steroid excess, however, the effects are deleterious. These are mediated through both direct and indirect effects. The latter results from decreased muscle mass, which plays a pivotal role in stabilizing joints and posture and thus decreasing fall risk [59]. The indirect effects consist of the stimulation of bone resorption through the increased activity of osteoclasts, which is further imbalanced by the induction of osteoblasts’ apoptosis and inactivity [50,60]. Increased bone resorption, further disrupted by replacing bone-forming cells with adipocytes, leads to bone mass reduction and a weakened structure [50,51,59]. In long-term users, up to 50% experience fractures due to both BMD loss and a decrease in bone strength linked to bone mineralization alterations. Even inhaled corticosteroids (ICS) might lead to lower BMD acquisition. Steroids impair cartilage growth through an interplay with growth hormone (GH) and IGF-1, and exhibit antiproliferative actions on chondrocytes and osteoblasts [59]. Steroid use, including ICS, has to be cautious in children because it has been found to impair their growth potential [61,62].

#### 3.3.2. Pathogenesis of ON in Corticosteroid Use

Steroid treatment with doses exceeding 20 mg daily has proven to be associated with an increased incidence of ON [55]. Nonetheless, even short-term, low-dose treatment is related to small yet statistically significant risk increases. In steroid-induced osteonecrosis, multiple pathomechanisms have been proposed over the last decades. The most discussed in the literature are [10,48,50,51,52,53,55,61,62,63,64,65]:Insufficient vascular supply and alterations in blood circulation;Inflammatory processes;Lipid metabolism alterations;Imbalance between osteogenesis and adipogenesis in the bone marrow.

Glucocorticoids might cause blood flow alterations through venal endothelial cell damage, which causes vasoconstriction and a hypercoagulable state [8,21]. Reduced venous blood flow results in stasis and increased intraosseous pressure [19,26,64]. Hypercoagulability, on the other hand, is related to thrombosis, which is accelerated through apoptosis of the endothelial cells and further reduces blood flow in small vessels [8,65]. A decrease in bone vasculature is a normal part of the aging process [50], but similar changes might also be observed with chronic steroid use [48,49,51]. Vascular supply develops primarily through vascular endothelial growth factor (VEGF) actions. In animals exposed to exogenous steroids, angiogenesis and blood vessel volume reduction were observed through the suppression of the transcription of both VEGF and hypoxia-inducible factor 1α (HIF1α) in osteocytes and osteoblasts [66]. However, in mice and in in vivo human studies, steroids decreased angiogenesis in the bone even in the presence of exogenous VEGF [66]. Notably, both angiogenesis and osteogenesis are correlated, as low VEGF expression negatively impacts osteoblasts and osteocyte differentiation and function. Cartilage and the growth plate experience similar detrimental effects of excess steroids through a VEGF-mediated mechanism, contributing to growth retardation in young patients [50].

In osteonecrosis, the inflammatory process, which usually is short-lasting and enables effective tissue repair, becomes chronic and destructive [43,67]. In steroid-induced ON macrophages, the cells responsible for clearing the osteonecrotic cells predominantly represent the M1 phenotype, which continuously excretes pro-inflammatory cytokines. An inflammatory state leads to the continuous death of bone cells and the exacerbation of tissue injury [43]. Preliminary findings suggest that agents which decrease the M1/M2 macrophage phenotype ratio decrease local inflammation and stimulate anti-inflammatory cytokine release, including of interleukin-10 (IL-10) and transforming growth factor-β (TGF-β), halting the inflammatory process [8,43,67]. 

Disorders of the lipid metabolism are one of the consequences of systemic corticosteroid use. Steroid-related dyslipidemia is characterized by a dose-dependent rise in total cholesterol and triglycerides, which have atherogenic properties [68,69]. It was hypothesized that hyperlipidemia has two actions that might provoke bone ischemia [8,19]. First, it promotes intravascular fat emboli formation. The second mechanism is increased pressure on the blood vessels and the bone structure itself, which results from fat accumulation in the bone marrow [8,55,70]. One study in rabbits, conducted by Zhao et al. [71], demonstrated the contrary, i.e., that a cholesterol-rich diet is associated with a reduced risk of ON in steroid-induced osteonecrosis. Of note, the study involved small groups (20 animals on a high-cholesterol diet and one with a standard diet). All animals received a single injection of 20 mg/kg methylprednisolone; thus, no control group was involved [71]. Therefore, their results need a cautionary interpretation before extrapolating into the effects of ON incidence in long-term steroid use in individuals eating a high-fat diet.

Mesenchymal stem cells in the bone marrow are pluripotential. Research on animals demonstrated that when steroids bind to their receptor in the bone marrow, they stimulate their differentiation into adipocytes through the peroxisome proliferator-activated receptor γ (PPARγ) and adipose-specific genes 422 (aP2) signaling pathways and transcription factors, such as coactivator transcription factor CCAAT/enhancer-binding protein α (C/EBPα) [63,72,73]. This results in hyperplasia of the fat cells in the bone marrow [18]. The decreased activation of the Cx43 gene, coding one of the connexins—proteins responsible for peak bone acquisition—was also proposed to play a role in steroid-induced ON [74,75]. Epigenetic changes affecting PPARγ might also increase the risk of ON [63].

Corticosteroid medications have opposing effects on cell proliferation and apoptosis depending on the type of cells and the dosage [7]. Thus, this explains the different effects of steroids on osteoclasts, osteoblasts, and osteocytes. As discussed earlier, steroids stimulate bone resorption through osteoclastogenesis while inhibiting osteoblastic lineage and osteocells’ proliferation, and increasing osteoblast autophagy, decreasing bone formation, resistance, and mineral density [7,19,21,50]. Alterations in osteoblastic activity and the prolongation of osteoclastic cells’ survival reduce bone resistance and mechanical strength [7,10,11]. Specifically, hyperactive osteoclasts’ activity during the bone regeneration phase is a major contributor to alterations in the bony structure. It is characterized by an interrupted bone structure, including bone loss and the subchondral trabeculae structure. That enhanced activity in steroid-induced ON is a result of free radicals formation and high levels of oxidative stress, which is not counteracted by scavenging mechanisms [63]. This results in a microscopic fracture which, accumulated over time, leads to ON.

When steroids act through glucocorticoid receptors (GR), they evoke an ani-inflammatory response through the activation of the nuclear factor κB, and numerous other interactions between nuclear co-modulators and transcription factors [7]. Though previous studies have recognized the imbalances between bone formation and resorption as the hallmark of osteonecrosis, new research also underlines the importance of chronic and uncontrolled inflammation, which results from an altered healing process after bone injury [43,67]. A prolonged inflammatory state hinders bone repair and regeneration, leading to ON [67].

Statins positively affect lipid metabolism and exhibit inhibitory effects on the PPARγ and HIF1α pathways involved in steroid-induced ON. Some research suggests that their use might be protective against steroid-induced osteonecrosis in humans, but more data are needed before recommending their routine use in ON prevention [63,72,76,77]. Table 3 summarizes the key information regarding alcohol- and steroid-induced ON.

#### 3.3.3. ON in Steroid-Dependent Conditions

Not all diseases are equal in terms of the risk of ON development following corticosteroid treatment. In several disorders, including systemic lupus erythematosus, renal and bone marrow transplant recipients, hematologic conditions, including Hodgkin’s lymphoma and acute lymphoblastic leukemia (ALL), and severe acute respiratory syndrome (SARS), there is an increased propensity to bone necrosis related to corticosteroids [52,53,54,55,56]. In fact, the first steroid-related ONFH case was described in a renal transplant recipient [51]. Although their use is sometimes necessary for numerous inflammatory conditions, the benefits do not always outweigh the risk of complications [52]. Genetic factors might also be involved in increased risk. As an example, Asian origin is related to an increased incidence of ON in steroid-dependent conditions [4,5,55].

In patients after renal transplantation, the incidence of ON is around 15% and is much higher in patients below the age of 35 (33% vs. 7%, *p* = 0.02) [57,78]. The higher the cumulative dose with every two-week interval during the months following organ transplant, the higher the risk of ON [79]. Newer treatment regimens have contributed to the decreased frequency of ON in renal transplant recipients over the last decades.

Hematologic malignancies themselves predispose to ON due to the increased risk of vascular complications related to microvascular leucostasis and thrombosis [80,81]. Most chemotherapeutic regimens used currently include a course of high-dose steroids to obtain and maintain disease remission, which further increases the risk [82]. In a study by Salem et al., 7.6% of children who developed AVN following ALL and non-Hodgkin’s lymphoma treatment received higher cumulative doses of steroids than patients without that condition (average 5.9 g vs. 3.9 g) [83]. Other studies suggest that the incidence rises with time, reaching 29% in the first decade after chemotherapy [84]. Due to high disability rates related to AVN in young age groups, screening for AVN might be therefore beneficial.

SLE is independently associated with a higher risk of ON [54]. Between 4 and 15% of patients with this condition will eventually develop symptomatic bone necrosis, but up to 40% have radiological evidence of ON [51]. Most (70–90%) SLE patients suffer from multifocal disease [85]. Many, however, require continuous steroid treatment, which allows them to remain in remission [81], and few alternative therapeutic options are being developed [10,81]. Each 2-month period of steroid therapy with high dose prednisone increases the risk of ON 1.2-fold (95% CI 1.1, 1.4) [86]. Gladman et al. demonstrated that cushingoid features, a common consequence of long-term systemic steroid use, increase the odds of ON 3.8 times in SLE patients [87].

The anti-inflammatory actions of steroids were used for the severe acute respiratory syndrome (SARS) to counteract excessive tissue damage due to pro-inflammatory cytokines’ overproduction. Nevertheless, their benefits were not clearly demonstrated [88]. Studies indicate that during the epidemic of SARS in 2003, more than 1 in 5 patients with SARS developed ON [21,55]. In a meta-analysis of studies conducted by Month et al., the relative risk (RR) for ON in patients receiving steroids was 1.57 with each 5 g increase in the cumulative dose (95% confidence interval (CI) 1.30–1.89, *p* < 0.001), and 1.29 (95% CI 1.09–1.53, *p* = 0.003) for every 10-day treatment period [52]. Some authors indicated that similar mechanisms might contribute to ON in COVID-19 patients receiving high-dose corticosteroids, but also that ON might be a part of post-COVID complications [88,89,90].

### 3.4. Strengths and Limitations

In undertaking this review, we employed a narrative approach to comprehensively synthesize the available knowledge on the links between osteonecrosis and steroid medication or alcohol consumption. A key strength of narrative reviews is their ability to provide a descriptive and critical analysis of diverse data sources without the rigid inclusion and exclusion criteria imposed by systematic reviews [8]. This inclusive approach allows for a broader exploration of the topic. However, it is important to acknowledge that narrative reviews may be susceptible to selection bias due to the less stringent criteria for source selection, potentially affecting the overall representativeness of the findings [91]. Furthermore, the interpretation and critical appraisal of studies by the authors conducting the narrative synthesis may introduce subjective elements that could impact the conclusions—but this type of error cannot be fully avoided in other types of reviews [92]. Additionally, narrative reviews may have limitations in terms of coverage, as the absence of strict search strategies and predefined criteria may result in unintentional omissions of relevant studies [92]. Furthermore, the lack of systematic methodologies, such as standardized search processes and transparent data extraction methods, introduces subjectivity and variability, potentially affecting the reliability and reproducibility of the findings. Despite these limitations, it is worth noting that narrative reviews are not arbitrary collections of data but are based on careful considerations of wide-ranging inclusion and exclusion criteria and are subject to the authors’ critical assessment.

## 4. Conclusions

Corticosteroids and alcohol use are the major risk factors for bone necrosis. When put together, both of them are the causative factors found in up to 80% of cases of ON [17]. For both agents, the risk is time- and dose-dependent [19,40,51,54,55]. Alcohol and steroid-induced ON share many pathogenetic mechanisms leading to the development of necrosis, including increased adipogenesis, the induction of chronic inflammation, vascular alterations, and impaired bone-cell differentiation [10,18,21,38,42,50,66].

According to a “multi-hit hypothesis’’ presented by Kenzora and Glimcher, a single risk factor is insufficient to trigger ON [93]. In this review, we have demonstrated that the risk of ON in an individual is further modified by a mix of other factors, including comorbidities, age, genetics, and others. Not all of these can be removed. Both alcohol and steroid use, however, are modifiable risk factors. Reducing exposure would contribute to the decrease in their overall impact on ON incidence and related disability. As the disease oftentimes affects multiple joints, in which ON develops independently [10,23,53], both general physicians and orthopedic surgeons should strongly advise their patients diagnosed with ON to limit their alcohol use, and encourage addiction treatment and make an intention to limit corticosteroid use to situations where these agents are essential.

## Figures and Tables

**Table 1 healthcare-11-01846-t001:** Keywords and group terms used in the literature search.

**Osteonecrosis**	OR	**Bone Health**	AND	**Alcohol**	OR	**Steroids**
Osteonecrosis Bone ischemia Avascular necrosis AVN Osteonecrosis of the femoral head ONFH		Bone health Bone impact Bone structure		Alcohol Alcohol abuse Alcohol impact Alcoholism Alcohol addiction ethanol		Steroid Corticosteroids Steroid dose Steroid treatment

**Table 2 healthcare-11-01846-t002:** Risk factors for osteonecrosis [1,2,3,5,10].

Internal Factors	External Factors
Diseases (lupus erythematosus, Gaucher’s disease, diabetes mellitus, and others) Hypercoagulability (e.g., in sickle cell disease, thrombophilias, and leukemias) Low mineral bone density Genetic factors Vascular abnormalities Metabolic alterations (hyperlipidemia) Hormonal imbalances (Cushing syndrome)	Medication (antiretrovirals, glucocorticoids) Alcohol abuse Trauma (physical trauma, decompression sickness, radiation)

**Table 3 healthcare-11-01846-t003:** Summary of study findings on alcohol- and steroid-related ON.

	Alcohol-Induced ON	Steroid-Induced ON
Epidemiology	Alcohol use is linked to 20–45% of ON cases [19,21] Risk is four times higher with regular consumption in comparison with occasional drinking (relative odds 13.1 vs. 3.2) [40] Risk increases with the amount of alcohol consumed [19]	Steroid treatment is found in 3–40% of patients with ON [19,20,21,51,52] Short-term high-dose therapy and chronic use are related to the highest risk [51,54,55]
Effect on bone health	Indirect [29]: Malnutrition, low body weight, sarcopenia [28,29] Micro- and macro elements’ deficiency (calcium, vitamin D, among others) Hormonal imbalances Direct: Decrease in osteoblast activity [29,34] Alterations in PTH secretion [34] Decreased vitamin D levels leading to ⇓ Ca^2+^ and P intestinal absorption [35] ⇓ in bone formation and mineralization [36,37]	Both excess and steroid deficiency impacts bone health negatively [51,59,60] Excess steroid use increases the risk of fracture indirectly (decreasing muscle strength, which leads to joint instability and falls) and directly (increased bone resorption) [51,59,60] Steroids stimulate osteoclasts, induce osteoblasts’ apoptosis and replace bone tissue with fat [50,51,60]
Pathogenetic mechanisms of osteonecrosis	Reduced BMD leads to microfractures and reduced remodeling [10,29] Increased adipogenesis in the bone marrow [38] Hypertrophic osteocytes compress the blood vessels and decrease blood supply, resulting in cell necrosis [18] Risk correlated with both duration and amount of alcohol [19,40]	Alterations of blood flow through venous stasis, thrombus formation and vasoconstriction, and blood vessel formation [10,19,66] Increase in inflammatory cytokines’ secretion by marcrophages [43] Steroid-induced dyslipidemia with atherogenic profile [68,69] Hyperplasia of bone marrow adipocytes [18]

BMD—bone mass density; Ca^2+^—calcium ions; ON—osteonecrosis; P—potassium; PTH—parathormone.

## Data Availability

Not applicable.

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
