# Peer review of "Osteonecrosis Related to Steroid and Alcohol Use—An Update on Pathogenesis"

_healthcare, 2023, doi:10.3390/healthcare11131846_

Round 1
Reviewer 1 Report (New Reviewer)
The paper regarding “Osteonecrosis related to steroid and alcohol use – an update on pathogenesis” is presented. The authors showed that osteonecrosis in alcohol and steroid share many pathogenetic mechanisms, including increased adipogenesis, induction of chronic inflammation, vascular alterations, and impaired bone-cell differentiation. The study is well-presented and interesting.
There are some possible issues
It was suggested (for example, PMID: 32309135) that osteonecrosis is regulated by multifactorial factors. It would be relevant to discuss this point.
Steroid uses would lead hyperactive osteoclasts ( for example, PMID: 32398957), and inflammation-induced osteoclasts. It would be informative to discuss the role of hyperactive osteoclasts and inflammation-induced osteoclasts in osteonecrosis?
In this paper, it was mentioned that angiogenesis was affected in osteonecrosis, one might wonder if the expression of bone angiogenic and angiocrine factors might contribute to this process, please discuss.
There are some typos, for example,
Because both alcohol and steroid useare ??
Table 1, consumptionSteroid? reducedAlterations?
Leucostasis?? Leukostasis (also called symptomatic hyperleukocytosis)??
There are some typos, for example,
Because both alcohol and steroid useare ??
Table 1, consumptionSteroid? reducedAlterations?
Leucostasis?? Leukostasis (also called symptomatic hyperleukocytosis)??
Author Response
Dear Reviewer.
Thank you for your positive feedback and comments. In the revised version of the article, comments have been made in the manuscript.
Comments:
It was suggested (for example, PMID: 32309135) that osteonecrosis is regulated by multifactorial factors. It would be relevant to discuss this point.
We have added a short paragraph at the beginning of the results section depicting how different risk factors contribute to ON occurrence. In addition, we have added a table that lists common risk factors for ON.
Steroid uses would lead hyperactive osteoclasts ( for example, PMID: 32398957), and inflammation-induced osteoclasts. It would be informative to discuss the role of hyperactive osteoclasts and inflammation-induced osteoclasts in osteonecrosis?
Thank you for your valuable instight. Please see the section 3.2.2., where we added a paragraph about the role of osteoclasts and oxidative stress in steroid-induced ON.
In this paper, it was mentioned that angiogenesis was affected in osteonecrosis, one might wonder if the expression of bone angiogenic and angiocrine factors might contribute to this process, please discuss.
Wer have briefly discussed it at the beginning of the 3.2.2. section.
There are some typos, for example,
Because both alcohol and steroid useare ??
Thank you for this remark. We have redacted the manuscript once again to make sure that there are no such mistakes. We hope that you are satisfied with the correction provided.
Table 1, consumptionSteroid? reducedAlterations?
Leucostasis?? Leukostasis (also called symptomatic hyperleukocytosis)??
Thank you for this remark. We have redacted the manuscript once again to make sure that there are no such mistakes. We hope that you are satisfied with the correction provided.
Reviewer 2 Report (New Reviewer)
The manuscript “Osteonecrosis related to steroid and alcohol use-an update on pathogenesis" by Wojciech Konarski et al aimed to summarize current knowledge on the two most prevalent factors leading to bone aseptic necrosis unrelated to trauma - steroid and ethanol-induced ON. The conclusion was that the risk of ON in an individual is further modified by a mix of other factors. Reducing exposure to alcohol and steroid use would contribute to the decrease of their overall impact on ON incidence and related disability.
The topic is interesting and the review is well written.
COMMENTS
1). The description of thiamine’s role is confusing, because the cited study uncovered the significance of B1 vitamin in bone health.
2). The role of vitamin D deficiency in the pathogenesis of osteonecrosis should be discussed more extensively.
3). The role of sarcopenia related to steroid and alcohol use should be better discussed.
c
Author Response
Dear Reviewer.
Thank you for your positive feedback and comments. In the revised version of the article, comments have been made in the manuscript.
COMMENTS
1). The description of thiamine’s role is confusing, because the cited study uncovered the significance of B1 vitamin in bone health.
Thank you for this remark; we have clarified how B1 deficiency may contribute to ON. In addition, we also mentioned that B2, which is also deficient in alcohol abusers, might also play a role in ON prevention.
2). The role of vitamin D deficiency in the pathogenesis of osteonecrosis should be discussed more extensively.
We have added a short description of its role in alcohol-related ON. Nonetheless, we want to keep our review focused on alcohol and steroids’ role in ON, therefore, we prefer to avoid an extensive discussion on that factor.
3). The role of sarcopenia related to steroid and alcohol use should be better discussed.
Though in both cases loss of muscle mass is present, this review is focused on osteonecrosis and the two risk factors. We have added a piece of information about sarcopenia in alcohol use and cirrhosis. Nonetheless, we want to keep our review focused on the main topic to avoid redundant digressions.
Reviewer 3 Report (New Reviewer)
The pathogenesis of ONFH is still unknown and further research is needed. Many readers will therefore be interested in how alcohol and steroids are related to the pathology of ONFH. I believe that this manuscript provides a brief description of alcohol- and steroid-related ONFH.
I don't think English quality is a problem.
Author Response
Dear Reviewer.
Thank you for your positive feedback and comments. In the revised version of the article, comments have been made in the manuscript.
Comments:
The pathogenesis of ONFH is still unknown and further research is needed. Many readers will therefore be interested in how alcohol and steroids are related to the pathology of ONFH. I believe that this manuscript provides a brief description of alcohol- and steroid-related ONFH.
I would like to express my heartfelt appreciation for your kind and encouraging feedback on the manuscript. I am sincerely grateful for your time and expertise in reviewing the manuscript. Thank you once again for your invaluable feedback. Your words of appreciation have inspired us to continue pursuing further research in this field.
Round 2
Reviewer 1 Report (New Reviewer)
This is a revised paper. The authors have addressed questions and the paper has been improved.
This manuscript is a resubmission of an earlier submission. The following is a list of the peer review reports and author responses from that submission.
Round 1
Reviewer 1 Report
This manuscript was well written and represents a good review paper with numerous references (total = 90). However, a few changes should be done as stated below:
1) In the abstract section, it is suggested to include the total number and range of years of the articles retrieved from the literature search.
2) On page 3 and line 99, I only noticed minor correction, please change "Is" to "is".
3) I would like to suggest the authors to include at least one Table to summarize the main findings.

Author Response
Dear Reviewer,
Thank you for your kind review on our paper. Your remarks have been adapted into the manuscript.
This manuscript was well written and represents a good review paper with numerous references (total = 90). However, a few changes should be done as stated below:
- In the abstract section, it is suggested to include the total number and range of yearsof the articles retrieved from the literature search.
We have added the total number of publication identified in the abstract section (line 21)
2) On page 3 and line 99, I only noticed minor correction, please change "Is" to "is".
That was corrected, thank you for noticing that error.
- I would like to suggest the authors to include at least one Table to summarize the main findings.
We have created a table with a brief summary of our findings, which compared both alcohol-related and steroid-induced ON (Table 1, page 8)
Best regards,
Reviewer 2 Report
This is a nice and comprehensive review of ON.
On 3.2 and on 3.2.3 the authors use the term OA, this is different than ON. They must correct this.
There is a MISLEADING approach regarding the incidence of ON of the femoral head and ON in other joints, humerus head, talus, in the use of steroids or alcohol. This must be commented WITH ETIOLOGY for the multilevel appearance of ON, like in leukemia, autoimmune disorders, alcohol and MAINLY steroid use.
LE is most commonly affecting the femoral head but ALL AUTOIMMUNE conditions affect ALL JOINTS.
Regarding the vascular supply, there are DIFFERENT vascular networks in the femoral head and the humeral head. This must be commented both in alocohol use and in steroid use.
The MANUSCRIPT is extensive, covers all aspects and MANY REMARKS can be brought because every paragraph contains a lot information
Author Response
Dear Reviewer,
We are thankful for your review of our paper. We have prepared a corrected manuscript based on your suggestion. Please find our responses to your comments below:
This is a nice and comprehensive review of ON.
On 3.2 and on 3.2.3 the authors use the term OA, this is different than ON. They must correct this.
Thank you for this comment, the headlines were corrected.
There is a MISLEADING approach regarding the incidence of ON of the femoral head and ON in other joints, humerus head, talus, in the use of steroids or alcohol. This must be commented WITH ETIOLOGY for the multilevel appearance of ON, like in leukemia, autoimmune disorders, alcohol and MAINLY steroid use.
LE is most commonly affecting the femoral head but ALL AUTOIMMUNE conditions affect ALL JOINTS.
Regarding the vascular supply, there are DIFFERENT vascular networks in the femoral head and the humeral head. This must be commented both in alocohol use and in steroid use.
Of course, there are differences in the vascularization of the femoral and humeral heads. There are also individual differences. But in the review, we did not discuss and develop this issue, because it is the basis for the next article (metanalysis). We hope that the mechanisms we have described are sufficient for the present article.
The MANUSCRIPT is extensive, covers all aspects and MANY REMARKS can be brought because every paragraph contains a lot information
Thank you for this feedback. We tried to make our review extensive in terms of information, but concise in form.
Best regards,
Reviewer 3 Report
1. In some sub-title, it seems that OA was confused with ON.
3.2. Epidemiology of OA in alcohol abuse
3.2.2. Pathogenesis of OA in corticosteroid use
3.2.3. OA in steroid-dependent conditions
2.Ggene polymorphisms have been found to modulate the risk of ethanol-induced ON.
Is there any difference among nations or race in these gene expression, just as the incidence of ON is different by nations or race
3. Physiological levels of endogenous steroids are necessary for bone mass maintenance.
However, excess endogenous and therapeutic steroids may seriously impact bone homeostasis.
Please explain the mechanism of this difference.
4. Steroid show anti-inflammatory effect in many disease. However, the steroid induces an proinflammatory effect in ON.
Please explain the mechanism of this difference.
5. Steroid used in systemic lupus erythematosus, renal and bone marrow transplant recipients, hematologic conditions,
including Hodgkin’s lymphoma and acute lymphoblastic leukemia (ALL), and severe acute respiratory syndrome (SARS).
Are there any stiudies comparing the risk and benefit of steroid in these disease?
Author Response
Dear Reviewer,
Thank you for your remarks on our paper. We hope that the corrections introduced to the manuscript and our responses will be satisfactory to you. Please find our responses below:
- In some sub-title, it seems that OA was confused with ON.
3.2. Epidemiology of OA in alcohol abuse
3.2.2. Pathogenesis of OA in corticosteroid use
3.2.3. OA in steroid-dependent conditions
Thank you for this remark, this is an obvious mistake. All headlines were corrected.
2.Ggene polymorphisms have been found to modulate the risk of ethanol-induced ON.
Is there any difference among nations or race in these gene expression, just as the incidence of ON is different by nations or race
The data on gene polymorphisms is scarce. We have added additional information from the references cited there. All of these studies were performed in Asian population. These authors refer to several other studies in their papers, but all are related to non-Caucasian populations.
- Physiological levels of endogenous steroids are necessary for bone mass maintenance.
However, excess endogenous and therapeutic steroids may seriously impact bone homeostasis.
Please explain the mechanism of this difference.
Both the physiological (the ‘good’) effects of steroids on the bone and the deleterious mechanisms of their excess are explained in brief in the lines 277-291.
- Steroid show anti-inflammatory effect in many disease. However, the steroid induces an proinflammatory effect in ON.
Please explain the mechanism of this difference.
The “proinflammatory” actions of sterioids in ON were explained in the section 3.2.2. (lines 316-324)
- Steroid used in systemic lupus erythematosus, renal and bone marrow transplant recipients, hematologic conditions, including Hodgkin’s lymphoma and acute lymphoblastic leukemia (ALL), and severe acute respiratory syndrome (SARS).
Are there any stiudies comparing the risk and benefit of steroid in these disease?
What you have highlighted here is a complex issue. There are many risks of steroid treatment and, at the same time – numerous benefits. ON is just one of the complications related to steroid use. We don’t think it would be feasible to conduct such a study as that would mean that a group of patients would not receive treatment, which in the case of hematologic disease would be rather unethical. Steroids have been incorporated into therapeutic regimens for lymphomas and leukemias for decades. Discussion over the benefits and risks of steroid use in some more chronic conditions, like SLE and other, is under debate. Such a discussion, however, is beyond the scope of this paper.
Best regards,